# Working Memory Training in Professional Football Players: A Small-Scale Descriptive Feasibility Study—The Importance of Personality, Psychological Well-Being, and Motivational Factors

**DOI:** 10.3390/sports7040089

**Published:** 2019-04-18

**Authors:** Dymphie In de Braek, Kay Deckers, Timo Kleinhesselink, Leonie Banning, Rudolf Ponds

**Affiliations:** 1Department of Neuropsychology, Maastricht University Medical Centre, 6229 ER Maastricht, The Netherlands; 2Department of Psychiatry and Neuropsychology, School for Mental Health and Neuroscience, Alzheimer Center Limburg, Maastricht University, 6229 ER Maastricht, The Netherlands; kay.deckers@maastrichtuniversity.nl (K.D.); leonie.banning@maastrichtuniversity.nl (L.B.); r.ponds@maastrichtuniversity.nl (R.P.); 3Hogeschool van Arnhem en Nijmegen, Sport en Bewegen, 6503 GL Nijmegen, The Netherlands; timokleinhesselink7@hotmail.com; 4Department of Medical Psychology, Maastricht University Medical Center, Maastricht and Adelante, Rehabilitation Center, 6432 CC Hoensbroek, The Netherlands

**Keywords:** working memory training, football players, feasibility

## Abstract

Background: Working memory training (WMT) programs can improve working memory (WM). In football players, this could lead to improved performance on the pitch. Method: Eighteen professional football players of Maatschappelijke Voetbal Vereniging Maastricht (MVV) participated and followed an online, computerized WMT program. Neuropsychological performance, psychological wellbeing, self-efficacy, and football skills (Loughborough Soccer Passing Test; LSPT) were assessed at three time points, before and after WMT and at three-month follow-up. Descriptive data are reported. Results: Baseline characteristics were roughly similar for both groups. Participants performed better on the trained WM tasks, but performance for other neuropsychological test measures or the LSPT did not change. Low compliance rates were observed, showing differences in personality and well-being between compliers and non-compliers. Conclusions: WMT is not a feasible and effective strategy to improve non-trained cognitive measures and football performance. However, this study indicates that it is important to take individual characteristics into account.

## 1. Introduction

Working memory (WM) is the cognitive skill that allows us to hold information in mind for just long enough to use it [1]. Information is held in mind as active neural traffic until it is forgotten, erased by the next job, or consolidated in long-term memory. WM tasks are those that require goal-oriented active monitoring or manipulation of information or behaviors in the face of interfering processes and distractions. WM can be seen as the “cognitive workbench” of the human brain, as it requires goal-oriented active monitoring or manipulation of information or behaviors in the face of interfering processes and distractions. The more people can station in their WM, the larger their cognitive ‘capacity’ and, as a consequence, their success in multi-tasking. WM—an important aspect of executive functioning (EF)—is associated with a wide range of complex cognitive behaviors, such as planning, impulse control, and reasoning [2,3], and is recently recognized as a predictor of success in top-football players [4]. WM forms the basis for learning, planning, linguistic, mathematical, and learning skills, organizing, staying focused, knowledge acquisition, impulse control, and reasoning [2,3]. EF includes the abilities of goal formation, planning, carrying out goal-directed plans, and effective performance [5]. EF skills are viewed as crucial developmental building blocks in cognitive and social abilities, with a focus on three core EF skills: working memory, inhibitory control, and cognitive or mental flexibility [6]. An excellent football player could be characterized by excellent cognitive functions, such as spatial attention, divided attention, inhibition, and WM. The player must be able to quickly adapt, change strategy, and inhibit responses in the ongoing, complex, and quickly changing stream of information during a soccer match. This is also referred to as “game intelligence” [4]. The paper of Vestberg et al. [4] suggests that both High Division (HD) and Lower Division (LD) football players had significantly better measures of executive functions in comparison to a norm group, for both men and women. Moreover, the HD players outperformed the LD players in these tests. In the second prospective part of the study, a partial correlation test showed a significant correlation between the result from the executive test and the numbers of assists and goals the players had scored two seasons later. The results from this study strongly suggest that cognitive test performance predicts the success of ball sport players [4].

WM can directly be trained by cognitive training programs, as has recently been shown in a variety of populations, for instance children with Attention-Deficit Hyperactivity Disorder (ADHD) and aging adults [7,8]. Klingberg et al. [9] developed a home-based computerized working memory training (WMT) program, resulting in better concentration, more control over impulsive behavior, and better complex reasoning skills. However, methodological shortcomings in many WMT studies have led to criticism on the use of WMT [6,10,11]. Only near-transfer, short-term improvements on verbal and nonverbal WM tasks were found, meaning that these effects are not generalizable to other functions or activities and are not sustained for a longer period of time. Even though there is critique on the WMT program, it is exciting to investigate the possibility of expanding WM capacity in football players through such program. If it is possible to improve the working memory function in football players, this could lead to improved performance on the pitch.

In the present study, we aimed to assess the feasibility of WMT in a group of professional football players. Primarily, the effect of WMT was investigated on trained WM tasks, non-trained WM tasks, and other cognitive measures. In addition, the generalizability of the effect of WMT to everyday life, in this case football performance, was investigated. In addition, non-cognitive measures, like self-efficacy and mental health, were included in this study. In previous research, it was shown that in professional sports, symptoms of common mental disorders, like distress, anxiety/depression, and sleep disturbance can negatively influence sport performance [12]. Since it was suspected that both psychological wellbeing and self-efficacy play an important role in WMT feasibility, these factors were examined as well. That is, self-efficacy induces people to strive, choose a certain activity, persevere and not give up, overcome temporary difficulties, and control the events that affect their lives, so that they can achieve their goals [13,14]. Also, the current literature suggests that competitive sport may contribute to poor mental health and that there is a stigma on mental health in athletes [15]. Finally, personality factors were included, as some, e.g., impulsivity, have been related to WM functioning [16].

## 2. Methods

### 2.1. Design and Population

Eighteen professional football players from Maatschappelijke Voetbal Vereniging Maastricht (MVV) in the southern part of The Netherlands participated in this study (all males; mean age = 23.78, SD = 4.08; range 17–31). All subjects gave their informed consent for inclusion before they participated in the study. At baseline, motivation to participate was evaluated via an interview, followed by a neuropsychological assessment. For this study, the WMT program Cogmed was used, supervised by a certified Cogmed coach. Directly after WMT and three months after, all players were assessed with the neuropsychological test measures and Loughborough Soccer Passing Test (LSPT). For this study a waitlist control design was used (see Figure 1). Subjects were randomly assigned to the experimental group (N = 8, mean age = 22.63, SD = 2.86, range = 20–29) or waitlist control group (N = 6, mean age = 25.00, SD = 5.40, range: 17–31).

### 2.2. Intervention

WMT (Cogmed) consisted of 25 computerized training sessions. Each session (duration of 30–45 min) consisted of a selection of various tasks that targeted the different aspects of WM. The participants were not directly trained on forwards or backwards digit recall, but on near-transfer tasks, e.g., memorizing which numbers lightened up on a screen. The training program was five-week-long with five sessions every week. The training was led by a qualified coach who worked with the user to provide structure, motivation, and feedback on the progress. The coach planned and structured the sessions with every individual, provided advice on how to get the most out of the training, gave personal feedback by mail every week on the training, and was available for questions. Further, if necessary, individual appointments intended to increase motivation were planned. However, no football players requested an additional appointment. Therefore, every football player received the same amount of attention/information. Both the user and the coach were able to review and monitor the results of each day’s training, using the online system. For this study, the coaches were two experienced neuropsychologists who completed the Cogmed coach program. The neuropsychological assessment was administered by trained psychologists.

### 2.3. Measurements

First, the neuropsychological assessment covered various cognitive domains. Intelligence was measured by the standard abbreviated form of the Groninger Intelligence Test-2 (GIT), consisting of several subtests: word list (woordenlijst), verbal abstract reasoning (matrijzen), puzzles (legkaarten), discovering of figures (figuur ontdekken), arithmetic (cijferen), fluency (professions and animals) [17]. Sufficient reliability, good concept validity, and sufficient criterion validity were reported for the overall GIT [18].

Short-term memory was assessed by the WAIS-III Digit span subtest (forward) and the Corsi Block-Tapping Task (CBTT); working memory was assessed by the WAIS-III Digit span subtest (backward) [19,20]. These tests were used as outcome measures of near-transfer effects. The digit span subtest of the WAIS consists of both a forward and a backward (reverse-order) recitation task, in which digit sequences are read aloud by an examiner. The sequences start with two digits that increase in length each trial. When subjects fail to accurately report back the sequence, testing is ceased [20]. Good reliability, sufficient concept validity, and insufficient criterion validity were reported for the overall WAIS-III [21]. During the CBTT, the participant had to mimic the researcher as he/she tapped a sequence of up to nine identical spatially separated blocks. It is a visual analogue of the WAIS-III Digit Span [22] which measures visuo-spatial short-term working memory. Because of the many variations and modifications of the task parameters, there is a paucity of assessments of its psychometric properties [23].

To evaluate learning capacity, efficiency of storage, and retrieval processes of newly learned verbal material, the Verbal Learning Test (VLT-15) was used [24]. Four parallel versions were used to avoid learning effects through rehearsal. In this test, 15 words were read out loud in a fixed order. After each trial, the patient was asked to reproduce the memorized words (immediate recall). Twenty minutes after the last trial, the patient was asked again to reproduce the set of words (delayed recall). Following the delayed recall, a list of 30 words was read out lout, and the subjects had to state whether or not the presented word was on the learning list (recognition). Acceptable reliability was reported for total and delayed recall scores [25].

The Stroop Color–Word Test (SCWT), the Concept Shifting Test (CST), and the Zoo map from the Behavioral Assessment of the Dysexecutive Syndrome (BADS) were used to measure EF, such as mental flexibility and vulnerability to interference [26,27,28]. In the SCWT, the interference of an automatic process (reading) with a more effort-demanding, controlled process task (naming colors) was investigated [27]. The SCWT consists of three subtasks: color word naming (I), color naming (II), and naming of color words printed in a different color (inference task III). The time needed to complete each card was scored [28]. The SCWT has been shown to have moderate to high reliability and validity [29]. The CST consists of three stimulus cards, and, on each test sheet, the participant is asked to cross out 16 circles in a certain order. All three stages are preceded by practice trails that contain six circles. In the first stage, CSTA, the trial is in numerical order, and in the second stage, CSTB, it is in alphabetical order. On the last stage, CSTC, both letters and numbers are shown, and circles need to be crossed out in number–letter–number–letter order, i.e., 1, A, 2, B, 3, C, etc. After this, a page with empty circles is shown (CST-0), and the participant has to cross them out, as fast as possible to compute the motor speed. Although many studies acknowledge the usefulness of the CST to assess concept shifting and related functions, its validity is still to be established [30]. The Zoo map subtest is a planning test that provides information about the ability to plan a route to visit 6 of possible 12 locations in a zoo. Reliability and validity have not been researched [31].

In addition to these objective instruments, the Cognitive Failure Questionnaire (CFQ; [32,33] and the Dutch version of the Working Memory Questionnaire (WMQ) were assessed [34]. Both questionnaires are self-report, with the CFQ consisting of 25 items (five-point scale, 0—never, 4—very often) measuring the frequency of everyday cognitive failures in the past 6 months, and the WMQ consisting of 30 items (7-point scale, 0—not at all, 5—extremely, 6—not relevant) measuring three dimensions of WM: short-term storage, attention, and executive control. Good test–retest and internal reliability of the CFQ were reported [35]. Good internal consistency and sensitivity were reported for the WMQ, and concurrent validity with the CFQ [34].

Secondly, differences in football skill performance in football players was assessed with the LSPT [36]. The LSPT requires players to complete 16 passes as quickly as possible, to listen to the test leader, memorize a color, decide where to pass the ball (four color options), and pass the ball. The time needed to complete 16 passes was used as the outcome measure. To improve accuracy, the LSPT was videotaped, and time was measured afterwards. The LSPT was reported to be a valid and reliable way to assess differences in football skill performance [36].

Thirdly, general psychological wellbeing was screened with the Symptom Check List-90 (SCL-90) [37], and self-efficacy, which is defined as the belief that one is capable of performing in a certain manner to attain certain goals, was assessed with the Dutch General Self-efficacy Scale (GSE) [38]. Personality was assessed with the NEO-Five-Factor Inventory (FFI), measuring five core personality traits (neuroticism, extraversion, openness, agreeableness, and conscientiousness) [39]. For the SCL-90, good internal consistency, construct validity, and divergent validity were reported [37]. For the GSE, high internal consistency and item-total correlations were reported, indicating that it is a reliable scale [40]. For the NEO-FFI, the foremost manner to test the validity is to replicate its factor structure, which the majority of studies have done, and the psychometric properties of the NEO-FFI are robust [41].

Finally, as for subject characteristics, age was used as a continuous variable. Educational level was indexed on an eight-point ordinal scale, ranging from primary to university education [24]. We used the “WMT start index” and “WMT index improvement” from the Cogmed software program as outcome measures of the effect on the trained aspects of WM. The WMT improvement index was calculated by subtracting the WMT start index from the WMT max index (i.e., mean of the three best trials of the two best training days).

### 2.4. Statistical Considerations

The neuropsychological test results of the complete football team were compared with the performance of healthy individuals of similar age and educational level by using normative data from the Maastricht Aging Study [42]. Descriptive statistics were performed to summarize sample characteristics. Given our small sample size, no statistical analyses were performed.

## 3. Results

The average WMT start index was 97.27 (SD = 8.87). Performance on the LSPT was 43.44 (SD = 2.07), which is comparable to earlier reported performance times of an elite football team (M = 43.6, SD = 3.8, [36]). The z-scores and c-scores of all neuropsychological tests indicated an overall average performance of this football team, reflecting cognitive performance within the normal range (Table 1). The experimental and waitlist control group showed to be roughly similar in terms of age, mean intelligent quotient (IQ) score, WMT start index, and performance on LPST (Table 1). Further, no clear change with regard to neuropsychological performance (Table 1), psychological well-being, subjective cognitive functioning, and self-efficacy was observed (Table 2).

### Feasibility

Of 18 football players who participated in this study, 14 started with WMT. Of these, 11 players completed WMT. Only 5 out of 14 players met the compliance criteria (defined as ≥20 training days within 5 weeks) [9]. Three subjects trained for 25 days, and two for 24 days. Overall, one subject did not meet the improvement index criterion (≥17), scoring 13.68. WMT improvement index ranged from 13.68 to 53.87. Of those who met the compliance criteria, the improvement index ranged from 31.75 to 53.87. Differences between compliers and non-compliers were observed for personality and well-being. For example, at t1, compliers scored lower on the NEO-FFI scale neuroticism (M = 21.67, SD = 1.52, versus M = 27.43, SD = 3.26) and higher on the scale extraversion (M = 52.33, SD = 3.21, versus M = 43.71, SD = 5.82) than non-compliers. Further, compliers scored lower on psychological wellbeing, i.e., psychoneuroticism, at t1 (M = 103.50, SD = 10.47, versus M = 114.33, SD = 15.20). We also observed drop-outs to be younger (M = 20.33, SD = 2.89 versus M = 24.55, SD = 4.03) and to have lower IQ scores (M = 83.67, SD = 22.75 versus M = 91.73, SD = 15.14) as compared to those who remained in the study.

## 4. Discussion

For the first time, this small-scale feasibility study evaluated the effects of WMT in professional football players. The WMT start index at baseline was similar for both the experimental and the control group. The WMT improvement index indicated that players performed better on the trained WM tasks (i.e., Cogmed task); however, no change on measures of neuropsychological tests, i.e., attention, verbal and visual WM, and planning, was observed over the study period, for both the experimental and the control group. Thus, we were not able to demonstrate a positive effect of WMT on non-trained neuropsychological measures, including measures of verbal and visual WM (i.e., the test battery as listed below Section 2.3). In addition, no change in football performance was found over the course of the WMT. Internal consistency with regard to LSPT scores was good, as mean LSPT scores found in this study resemble those reported in prior studies that assessed soccer performance in adolescents and young elite players [36]. It is important to note that at baseline, the experimental and control group were similar with respect to demographic variables (e.g., age, level of education, or measures of intelligence), personality, psychiatric symptoms, subjective cognitive functioning, and self-efficacy.

Interestingly, we observed that non-compliers were emotionally more instable and experienced more psychological complaints than compliers. This seems to indicate that the individuals who experienced less (cognitive) problems were more willing and able to commit to the WMT program. Also, it has to be noted that drop-outs were on average younger and had a lower IQ score.

A strength of the current study is that it was conducted in a natural setting and used a random assignment of football players to the different groups. The major limitation of the current study is the small sample size, because of which we were not able to follow through with our planned statistical testing. The most important observation made during the study was a lack of motivation of the football players. Only 5 out of 14 players met the compliance criteria (defined as ≥20 training days within 5 weeks). This was disappointing and somewhat unexpected, as participation was stimulated by trainers and medical staff of MVV. Furthermore, the coaches were present every week for questions, problems, and to improve motivation and were available for questions through mail. However, previous studies in the adult population also encountered difficulties with recruitment and retention, such as in subjects with ADHD [43]. High attrition rates were attributed to the fact that the WMT did not meet subjects’ levels of acceptability or practicality. Further, the program was experienced as far more time-consuming than adverted [43]. The authors reasoned that the difference with respect to the efficacy rates seen in children might be explained by the fact that the latter were closely monitored and scaffolded by clinicians and caregivers [43]. We hypothesize that football players did not experience enough cognitive complaints as shown by the results of the CFQ. In daily life practice, it seemed difficult for football players to make the translation between cognitive exercises and performance on the pitch, since there was no direct reward. This could indicate that low internal motivation was present for intensive WMT, and, therefore, the need to incorporate the WMT in their daily routine was not recognized as such. It is important to note that our study was initially meant as a small-scale feasibility study, and the fact that we were able to examine factors that we hypothesized to influence compliance rates (e.g., personality traits) in an objective manner can therefore be considered as a strength.

The main point of criticism regarding WMT is that improvements (if at all) are not found to be generalizable to other functions or activities [8,9]. It is worth noting that improvements in memory tasks after intensive practice could be the result of task-specific strategies, not of the improvement of memory itself. Some even argued that WMT programs are based on a naïve “physical–energetic” model: training WM (process, e.g., such a exercising a muscle) will not necessarily result in improvement in that process (e.g., strengthening that muscle) [10].

## 5. Clinical Implications

The results of this small study show that WMT is not a feasible and effective strategy to improve WM and football performance. Further, it is essential to characterize the psychological wellbeing of the subjects before starting an intensive computerized WMT. In this study, we did not focus on individual patterns of WMT benefits. It is likely that some football players will benefit more than others from WMT. Future research should investigate individual patterns, like personality, cognitive failures, and psychological wellbeing and their influence on WMT programs.

## Figures and Tables

**Figure 1 sports-07-00089-f001:**
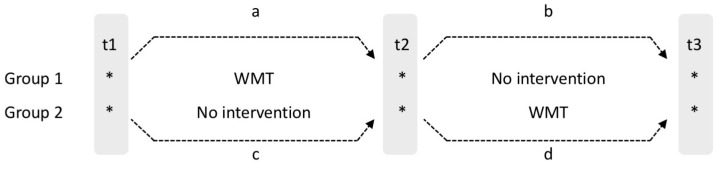
Study Design. a, b, c and d indicate changes in outcome variables; with a and d indicating an intervention effect; in case a significant effect on a is found, these can be compared to b, with the direction of change indicating whether there is an intervention effect (i.e., no change or positive change) or a temporary effect (negative change).

**Table 1 sports-07-00089-t001:** Subject characteristics pre- and post-working memory training (WMT).

	Standardized Scores *	Unstandardized Scores **				
	Total Group		Experimental Group		Waitlist Control	
	Pre-WMT	Pre-WMT	Pre-WMT (t1)	Post-WMT (t2)	Pre-WMT (t2)	Post-WMT (t5)
	(N = 18)	(N = 18)	(N = 8)	(N = 5)	(N = 6)	(N = 6)
Age		23.78 (4.08)	22.63 (2.83)	NA	25.00 (5.40)	NA
Cogmed start index		97.27 (8.87)	97.18 (10.01)	NA	97.38 (8.12)	NA
LSPT		43.44 (2.07)	42.93 (2.30)	39.88 (4.64)	40.66 (0.68)	38.20 (2.67)
*GIT*						
IQ		89.06 (16.63)	86.25 (12.03)	NA	95.00 (21.00)	NA
Wordlist	4.33 (2.52)	11.00 (3.82)	11.00 (3.42)	NA	11.50 (3.39)	NA
Puzzles	4.67 (2.09)	12.56 (3.75)	11.63 (3.81)	NA	13.33 (4.27)	NA
Figures	4.17 (3.19)	11.44 (5.09)	9.87 (4.76)	NA	11.50 (6.38)	NA
Arithmic	4.72 (2.27)	8.56 (3.28)	9.63 (2.62)	NA	9.33 (3.88)	NA
Verbal Abstract Reasoning	4.00 (1.57)	11.61 (2.38)	11.13 (2.17)	NA	12.33 (3.01)	NA
Fluency – Animals	3.83 (2.12)	22.50 (6.20)	22.13 (7.79)	NA	24.33 (5.35)	NA
*SCWT*						
Interference	0.42 (0.81)	32.06 (11.04)	29.57 (2.56)	31.36 (5.08)	24.45 (8.11)	22.63 (6.86)
CST						
Interference	-0.15 (1.07)	6.99 (5.96)	5.96 (3.32)	6.50 (3.85)	5.32 (3.37)	8.94 (10.80)
WAIS-III digit span						
Forward		9.56 (2.04)	9.88 (1.46)	11.60 (1.82)	10.50 (2.88)	11.33 (3.56)
Backward		6.83 (2.66)	7.13 (2.59)	9.60 (2.19)	10.00 (2.28)	10.33 (2.87)
*Corsi*						
Total score		9.78 (1.59)	10.50 (1.07)	10.60 (1.67)	10.00 (1.10)	11.00 (2.45)
Longest sequence		6.39 (0.78)	6.75 (0.46)	6.40 (0.89)	6.50 (0.55)	6.50 (1.22)
*BADS Zoo*		3.17 (0.92)	3.13 (0.64)	3.00 (1.00)	3.00 (1.26)	3.83 (0.41)
*VLT*						
Immediate Recall	0.99 (1.06)	57.44 (8.03)	57.00 (8.81)	64.20 (6.91)	64.50 (4.46)	66.33 (4.13)
Delayed Recall	0.74 (0.83)	12.83 (1.98)	12.88 (1.89)	13.60 (1.67)	14.17 (1.17)	13.83 (1.83)

*Note.* LSPT is Loughborough Soccer Passing Test; GIT is Groninger Intelligence Test-2; IQ is intelligence quotient; SCWT is Stroop Color–Word Test; CST is Concept Shifting Test; WAIS-III is Wechsler Adult Intelligence Scale; Corsi is Corsi block-tapping task; BADS zoo is Behavioural Assessment of the Dysexecutive Syndrome, Zoo map; VLT is Verbal Learning Test-15. ** LSPT Experimental group (N = 5) and Waitlist control (N = 5). * standardized scores, i.e., computed c-scores for the GIT and computed z-scores (using normative means) for the SCWT, CST, and VLT; ** unstandardized scores, i.e., raw test scores.

**Table 2 sports-07-00089-t002:** Baseline measures of personality, psychiatric symptoms, subjective cognitive functioning, and self-efficacy.

	Total Group	Experimental Group	Waitlist Control
	M (SD)	M (SD)	M (SD)
*NEO-FFI*	N = 13	N = 7	N = 3
Neuroticism	25.92 (3.97)	27.00 (3.65)	22.67 (3.06)
Extraversion	45.69 (6.33)	43.86 (5.98)	52.00 (3.61)
Openness to Experience	30.54 (4.68)	28.86 (5.08)	33.67 (3.51)
Agreeableness	44.39 (5.49)	42.00 (4.73)	48.67 (7.57)
Conscientiousness	48.08 (5.66)	48.57 (3.55)	52.00 (8.89)
*SCL-90*	N = 11	N = 6	N = 4
Psychoneuroticism	109.82 (13.30)	112.83 (14.48)	105.75 (14.10)
	N = 11	N = 6	N = 4
*GSE*	30.46 (3.21)	29.17 (3.19)	33.00 (1.83)
*CFQ*	69.91 (12.26)	70.33 (11.76)	75.50 (4.65)
*WMQ*	48.73 (12.92)	49.00 (14.95)	11.00 (7.35)

*Note.* NEO-FFI is NEO Five-Factor Inventory; SCL-90 is Symptom Check List—90; GSE is Dutch General Self-Efficacy Scale; CFQ is Cognitive Failures Questionnaire; WMQ is Working Memory Questionnaire. Data are presented as raw test scores.

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
