# Peer review of "Working Memory Training in Professional Football Players: A Small-Scale Descriptive Feasibility Study—The Importance of Personality, Psychological Well-Being, and Motivational Factors"

_sports, 2019, doi:10.3390/sports7040089_

Round 1
Reviewer 1 Report
The manuscript describes an interesting study concerned with the effects of working memory training on cognitive functions as well as football performance. The findings would be of interest to readers of this journal, and the manuscript is also well-written. However, I am unable to recommend the manuscript for publication in its current form. I provide justification below:
The manuscript would be better as a lengthier article, providing more theoretical background. For example, there is a lack of clarity surrounding the underlying structure and functions of executive functions (page 1). It is suggested that working memory is associated with cognitive behaviours such as planning. However, many researchers would consider planning to be an executive function in itself, and not a behaviour per se. The references provided in this part of the introduction are also dated, and not the most suitable references for defining and describing the function of executive functions. There could also be more detail about which aspect(s) of executive function or working memory may play a role in football.
The introduction (page 2) also needs to be developed further in terms of why personality, wellbeing, and self-efficacy were measured. Why was it suspected that these may play a role? What previous research informs this?
There needs to be more detail in the method section. The number of participants should be described in terms of the two groups separately (i.e. rather than 18 participants Table 1 notes there were 8 in an intervention group and 6 in a control). There also needs to be some detail about what each of the cognitive assessments involved, and a comment about their reliability.
The results could be presented more clearly e.g. the Tables how baseline descriptives but it would be more useful if they displayed scores for both pre and post intervention. For completeness should Table 2 not also show results for the working memory questionnaire?
In the interpretation of the results and their discussion there needs to be more clarity about what is considered a transfer task. So were participants directly trained on forwards digit recall (which improved) but not backwards recall (which did improve)? The discussion could also do a better job of integrating the findings from and inclusion of the cognitive and non-cognitive measures.
Finally, I also have some concerns about the sample size of the study. If the other issues raised were addressed then I would perhaps be less concerned, but a more detailed discussion about this could be included (maybe making reference to previous studies with similar sample sizes or perhaps emphasising that this is a small scale feasibility study).
Author Response
See the uploaded word document

Reviewer 2 Report
Review „Working memory training in professional football players: a small feasibility study”
Summary
The authors present a feasibility study for working memory training with professional football players. They apply working memory training over a period of five weeks and observe no significant transfer to any other neuropsychological test measures the passing test they applied. I like the idea and I believe that the study is generally suitable for the journal but I have several fundamental concerns. The sample size is extremely small, so significance testing does not make much sense to me, especially considering the large amount of tests applied in this study. Moreover, throughout the article, the authors don’t explain their choice of tests (why would they test personality traits three times?). Therefore, I believe that less is more in this case: The authors should clearly state why they used the tests they applied, formulate hypotheses and use an adequate statistical model (if they insist on testing for statistical significance – I think that descriptive statistics would do for such a small sample), as described below. In line with this, I believe that given the small sample size and the resulting low statistical power, the major merit of this paper lies within the discussion of the influence of compliance and related factors (at least for the current manuscript version). This should therefore be emphasized and discussed more thoroughly.
Major
· A mixed model would be better suited for the analysis, especially because a third measurement point can be taken into account. Also, no need to worry about baseline differences anymore (as you do on P9). Here is an article analyzing working memory training transfer on intelligence with exactly such model:
Hilbert, S., Schwaighofer, M., Zech, A., Sarubin, N., Arendasy, M., & Bühner, M. (2017). Working memory tasks train working memory but not reasoning: A material-and operation-specific investigation of transfer from working memory practice. Intelligence, 61, 102-114.
Obviously, I am one of the authors, so no need to cite this paper but I believe a mixed model for longitudinal training data with experimental groups is much more suitable for your study compared to the myriad of t-tests you perform (without correcting for multiple testing).
· Why would you test for significant differences in personality traits? As the name implies, they are traits, so what’s the reasoning behind testing if the test results for the five factors differ between the time points? If you’re not looking to find an answer for a research question, I suggest you refrain from significance testing.
Minor
· Why was there no correction of type-1-error probability? You list it as a limitation but why didn’t you simply do it?
· Individual appointments to increase motivation are a possible confound. Did you keep track of the appointments of the individual players? This would be important information for a feasibility study.
· Digit span forward is not t a working memory test but a short term memory test
· P7: What Z-scores? The table depict various scores and none of them are labeled as Z-scores. Given that Z-scores yield an average of M = 100, I assume that you refer to Psychoneuroticism? Anyway, this should be labeled adequately in the legend.
· P7: Where is the IQ depicted? Can it be that you confused Table 2 with Table 1 (maybe this also holds for my previous issue with the Z-scores)?
· P7: Which of the personality scores differed between the compliant and non-compliant subjects? The ones at time point 1? At another time point? An average score?
· P7: Extraversion is not a subscale of the NEO-FFI, it is one of the scales. There are subscales of extraversion in the NEO-PIR, but that’s a different story
· How does the finding that a non-significant difference in extraversion between two groups that you (randomly) assigned suggest that – in the light of null-findings – “individual differences might influence efficacy of WMT”. I believe this is far-fetched. Don’t get me wrong, it may well be, but this seems pretty unsound statement based on the findings you report.
· If you argue with statistical power (P9), you should report it.
Author Response
See the uploaded word document

Round 2
Reviewer 1 Report
I reviewed the original submission, and have since read the response to my earlier comments and the revised manuscript. The revisions have improved the manuscript substantially, particularly in terms of the theoretical background to working memory and executive functions, and also the reporting of the methodology. However, some of the comments have only been addressed briefly and thus there is still room for improvement.
My main concern here is the justification for the non-cognitive measures (personality, wellbeing, motivational factors). There is nothing at all in the introduction to justify measuring personality. In the response to the reviewers comments it suggests that personality has been removed, but it is still in the method and results sections. The justification for measuring wellbeing is also weak- why would competitive sport leading to poor mental health justify measuring it here? There needs to be more background literature about this claim as well.
Reviewer 2 Report
Review 2 „Working memory training in professional football players: a small feasibility study”
The authors followed some of my minor suggestions and adapted the manuscript accordingly. However, conducting and interpreting a great number of t-tests with group sizes of six and eight simply does not make sense. I would have accepted a more robust overall model without significance testing but the authors decided to stick to significance testing with their large amount of comparisons and their tiny sample. They, of course, have the right to do so, yet I strongly believe that this does not make sense from a scientific point of view, so I have to suggest a rejection for the paper in its current form. Testing for significant differences in psychological wellbeing in two randomly assigned groups at t1 – especially given the sample size – is simply not what significant tests were invented for. Taken together, I like the idea of the study and I think it would be worth publishing as a descriptive feasibility study but it does, in my opinion, in its current form, simply not meet the quality standards of a scientific journal. I listed some minor suggestions to help improving the manuscript, which are listed below.
Minor suggestions
· Now you report the digit span forward test and the digit span backwards test. The first one is a short term memory test, the second one a working memory test. This is why the two versions exist: to measure these two different constructs.
· What standardized scores? z-scores (unlike Z-scores) yield an average of M = 0. So the results you report are pretty high. Also, why the ** for LSPT?
· P8: Typically, it’s p £05 instead of p < .05 (even though it’s a point estimate for a continuous variable, so it does not make a difference on most levels)
Round 3
Reviewer 2 Report
The authors followed my suggestion and refrained from significance testing but rather described the data adequately and focused on feasibility. This approach is scientifically sound as well as informative for the reader. I appreciate the effort they put into removing the significance tests and changing the focus of the manuscript. I think the manuscript is publishable in its current form.